# The Correlation of Papanicolaou Smears and Clinical Features to Identify the Common Risk Factors for Cervical Cancer: A Retrospective and Descriptive Study from a Tertiary Care Hospital in Trinidad

**DOI:** 10.3390/vaccines11030697

**Published:** 2023-03-18

**Authors:** Srikanth Umakanthan, Maryann M. Bukelo, Saudah Ghany, La Donna Gay, Tia Gilkes, Jamila Freeman, Andre Francis, Kyle Francis, Gabriel Gajadhar, Junea Fraser

**Affiliations:** 1Pathology Unit, Department of Paraclinical Sciences, Faculty of Medical Sciences, The University of the West Indies, St. Augustine 685509, Trinidad and Tobago; 2Department of Clinical Medical Sciences, Faculty of Medical Sciences, The University of the West Indies, St. Augustine 685509, Trinidad and Tobago

**Keywords:** Papanicolaou smears, epithelial cell abnormality, cervical cancer

## Abstract

**Background:** Cervical cancer, the fourth most frequent cancer in women, is associated with the human papillomavirus (HPV). This study identifies risk factors and clinical findings for abnormal cervical cytology and histopathology in the Trinidad and Tobago populations. Some risk factors include early age of first coitus, a high number of sexual partners, high parity, smoking, and using certain medications, such as oral contraception. This study aims to identify the significance of Papanicolaou (pap) smears and the common risk factors that contribute to the development of premalignant and malignant cervical lesions. **Method:** A three-year retrospective, descriptive study of cervical cancer was conducted at the Eric Williams Medical Sciences Complex. The subject population included 215 female patients aged 18 years and older with the following documented abnormal cervical cytology: (ASCUS), ASC-H, LSIL, HSIL, Atypical Glandular cells, HPV, Adenocarcinoma, and Invasive Squamous Cell Carcinoma. Histopathology records were analysed for thirty-three of these patients. Patients’ information was recorded on data collection sheets adapted from the North Central Regional Health Authority’s cytology laboratory standardised reporting format request form. **Results and Findings:** The data were analysed via Statistical Package for Social Sciences (SPSS) software edition 23 using frequency tables and descriptive analysis. The mean sample age of the population was 36.7 years, the first age of coitus was 18.1 years, the number of sexual partners was 3.8, and the number of live births was 2. LSIL was the most popular abnormal finding, 32.6%, followed by HSIL, 28.8%, and ASCUS, 27.4%. Most histopathological reports resulted in CIN I and II. **Conclusions**: The significant risk factors observed for cytology abnormalities and premalignant lesions were early age of coitus, a high number of sexual partners, and no use of contraception. Patients mostly presented as asymptomatic despite obtaining abnormal cytology results. Hence, regular pap smear screening should continue to be highly encouraged.

## 1. Background

Cervical cancer is ranked as the fourth most common female cancer worldwide [1]. It is also estimated that most deaths (90%) occurred in low- and middle-income nations. In the Caribbean, it is the second cause of death among females [2]. Its mortality rate is thrice that of North America and is estimated to increase by 45% by the year 2030 [3]. Taking this into account, screening programmes are at the forefront of early detection of patients who may develop cancer. Cervical screening guidelines for the Caribbean region recommend that, at a minimum, 80% of women aged 25–50 years who have ever engaged in sexual intercourse should be screened for precancerous cervical lesions. Additionally, women with abnormal cervical smear results should have repeat cervical screening at consecutive intervals or gynaecological evaluation [4].

Malignancy of the cervix is mainly of two histological types: the more common squamous cell carcinoma and a rarer form called adenocarcinoma. There are two types of screening programmes: opportunistic and comprehensive/organised screening programmes. With opportunistic screening, patients request a test to be performed for any reason, whilst with comprehensive, large groups of women are persuaded to have the test performed to identify those who may be asymptomatic.

Trinidad and Tobago were ranked seventh in the Caribbean for mortality rates of cervical cancer (10.8 per 100,000) and, although there has been an increase in screening programmes in Trinidad and Tobago, there has not been a decrease in mortality rates. This can be attributed to the fact that Trinidad and Tobago do not have a formal cervical cancer screening programme that is uniformly applied; instead, each regional health authority (RHA) is responsible for its own screening programme.

Cervical cell abnormalities include squamous cell and glandular cell abnormalities. Squamous cell abnormalities include low-grade squamous intraepithelial lesions (LSIL), high-grade squamous intraepithelial lesions (HSIL), atypical squamous cells of undetermined significance (ASCUS), atypical squamous cells cannot exclude HSIL (ASC-H), and invasive squamous cell carcinoma. Glandular cell abnormalities include atypical glandular cells and adenocarcinoma [5]. These cellular abnormalities are identified in satisfactory specimens obtained from cervical samples during screening programmes or following symptomatic clinical evaluation of cervical pathology diagnosis [6].

In Trinidad and Tobago, the screening programmes are conducted mainly to identify early premalignant cervical lesions through Papanicolaou (pap) smears in females older than 18 years and continue until the age of 65 years [2]. Since cervical cancer is mostly caused by Human Papilloma Virus (HPV), a sexually transmitted disease, the screening programmes usually target populations with low socio-economic regions [7]. Patients with comorbidities, positive family history of cervical and related cancers, unsafe sexual practices, and females with symptomatic cervical lesions add to the high-risk categories [8].

The aims of our study were as follows: 1. to identify premalignant and malignant cervical neoplasms using pap smears; 2. to correlate its significance with identifiable risk factors and clinical features; and 3. to correlate the cytological findings with histopathology diagnosis.

This study further provides and establishes the significance of pap smears in screening programmes and aids in efforts for prevention, early diagnosis, and successful management of cervical cancers.

## 2. Methods

This retrospective and descriptive study comprised the screening of 6208 pap smears using standard cervical cytology protocol over three years. The selection criteria consisted of patients who underwent pap smear examinations at Mt. Hope Women’s Hospital (MHWH) and health centres of the NCRHA and had abnormal cytology results. Among these, 215 cases were identified with cervical cytology abnormalities. They were further evaluated for patient clinical and histopathology information from the medical records department, which included patient demographics, risk factors, medical and sexual history, family history, clinical findings, and histopathology diagnosis. Histopathological correlations were performed on 33 traceable cases. The data obtained were analysed with the descriptive data analysis Statistical Package for the Social Sciences (SPSS) version 21 Premium. Descriptive data analysis included calculations of central tendencies’ means, medians, modes, ranges, and other estimates of the demographic of the sample generated. Further analysis using frequencies and percentages was utilised to determine the risk factors of abnormal cervical cytology and histopathology.

## 3. Results

As seen in Table 1, 66 patients (30.7%) of the sample population came from 34–41 years old. Most of the patients, 136 (63.3%), were Afro-Trinidadians; 33 (15.3%) were Indo-Trinidadians; and 46 (21.4%) were of mixed ethnicity. Most patients in the population were single, 92 (42.8%). Most patients hailed from Eastern Trinidad, 98 (45.6%), and Northern Trinidad, 77 (35.8%).

Table 2 shows notable risk factors: the mean sample age for first sexual intercourse was 18.1 +/− 3.2 years, the minimum age was ten years, and the maximum was 30 years. The sample mean for sexual partners was 3.8 +/− 3.5, the minimum was one partner, and the maximum was 23 partners. The sample means for the number of live births was 2.0 +/− 1.8, while the mean sample number of pregnancies was 2.9 +/− 2.5.

Most patients, 132 (61.4%), did not have any noted medical history. However, 83 (38.6%) had specific comorbidities or illness. 25 (11.6%) had hypertension, 11 (5.1%) had diabetes, and 11 (5.1% had HIV). For HIV patients, the number of sexual partners ranged from 2–23, and the first age of coitus ranged from 10–20 years old. Other noteworthy mentions are that 17 patients had asthma, 6 had lupus, and 2 had a history of cervical cancer. Fourteen patients (6.5%) also admitted to smoking.

Table 3 shows that 133 (61.5%) did not have a family member with cancer, while 82 (38.5%) did. Of those who did, 6.5% of family members had cervical cancer, of which were primarily secondary relations; 14% had another type of female reproductive cancer such as breast, ovarian, and uterine; and 79.5% had other types of cancer, such as stomach, pancreatic, prostate, and lung. Table 4 elaborates that most patients did not use any form of contraceptive, 133 (61.9%), while 82 (38.1%) used.

From Table 5, most patients exhibited normal bleeding, 177 (82.3%), while 38 (17.7%) had abnormal bleeding. Most patients, 147 (68.4%), had no discharge. The clinical appearance of the vagina and vulva was normal for most patients, 214 (99.5%) and 213 (99.1%), respectively. The cervix was normal for 170 (79.1%) patients; 36 (16.7%) had contact bleeding, 3 (1.4%) had erosion, 3 (1.4%) had chronic cervicitis, and 1.4% had a polyp. These were incidental clinical findings during the screening program.

Table 6 shows that the most common cytological finding was LSIL in 70 (32.6%) of patients. This was followed by HSIL, seen in 62 (28.8%) of patients, and ASCUS, seen in 59 (27.4%). Eleven (5.1%) cases with HPV and no Adenocarcinomas were found.

Out of the 33 patients whose histopathology reports were accessible. Table 7 elaborates that 12 (36.4%) of these patients had CIN I, 11 (33.3%) had CIN II, 9 (27.3%) had CIN III, and there was one case (3.0%) of Invasive Squamous Cell Carcinoma (SCC) reported.

## 4. Discussion

Literature studies in the Caribbean that examine the top ten causes of cancer death (5-year cumulative proportions) among women in 21 Caribbean countries found that cervical (11%), endometrial (6.4%), and ovarian (5.2%) cancers were ranked 2nd, 4th, and 6th, respectively. In terms of cervical cancer mortality, rates were highest in St. Vincent and the Grenadines (15.5 per 100,000), Belize (12.5 per 100,000), and St. Kitts and Nevis (12.3 per 100,000), while Trinidad and Tobago (TT) had the 7th highest rate (10.8 per 100,000). Strikingly, cervical cancer mortality rates in the United States (US), US Virgin Islands, and Puerto Rico, which have well-established cervical cancer screening programs as part of the Center for Disease Control and Prevention’s National Breast and Cervical Cancer Early Detection Program, were approximately seven times lower than the rate for TT. Our recent analysis of TT cancer surveillance data (unpublished data) showed that the cervical cancer mortality rate in TT was 9.7 per 100,000. There are few studies on cervical cancer and its risk factors in the Caribbean. Some of these risk factors are from exposure to Human Papilloma Virus (HPV), specifically types 16 and 18, older age at diagnosis, decrease in accessibility to contraception, first sexual encounter before 20 years of age, high parity, smoking, oral contraceptive pill usage exceeding five years, family history of cervical cancer and immune-depression, and systemic disease [2,4,6,9].

In our study, most patients were between 18 and 41 years old, coinciding with a previous report of 15–44 years [10]. Afro-Trinidadians were the majority, representing 63.6% of the sample population. This trend was also seen in the United States of America, where African American and Latina populations could not afford routine pap smear screening [11]. However, in Trinidad, pap smears are free of charge at all health centres and some hospitals. Therefore, a possible reason for the high number of Afro-Trinidadians in this study may be due to the region of sampling as our sample site captured the population from the Eastern and Northern areas of Trinidad, where the Afro-Trinidadian population is high.

Findings in our study revealed that most of the women were single (42.8%). It is, therefore, more likely that these individuals would have had more sexual partners when compared to their married and common-law counterparts, hence an increased risk of contracting HPV.

There are no conclusive links between medical history and cervical cancer. The high number of hypertensive and diabetic patients in our sample can be attributed to the fact that they are Trinidad’s most prevalent lifestyle diseases.

However, Blumenfeld (1994) found that women with systemic lupus erythematosus (SLE) had higher incidences of cervical pathologies when compared to non-SLE women [12]. Six SLE cases were found in our study, and 5.1% of our sample population were HIV positive. An immune-deficient system is less efficient in protecting the body from HPV and is more susceptible to cell abnormalities. HPV, a sexually transmitted disease, is strongly linked to cervical cancer. One study (Burd, 2003) even noted that such an association was more significant than the association between smoking and lung cancer. HPV types 16 and 18 are known for predominantly infecting the cervix resulting in atypical growth of cells, and it has been found in almost all (99.7%) cervical cancer cases [13].

No significant association existed between one’s family medical history of cervical cancer and a woman’s predisposition to acquire the disease. This disease, like many other cancers, is multifactorial, and it should be noted that lifestyle choices have a more significant influence. However, familial aggregation is possible from genetic susceptibility, shared environmental exposure, and shared lifestyle practices among family members [14]. Genetic susceptibility includes genes that make patients more vulnerable to persistent HPV infections, including those that regulate immunity and susceptibility [15]. Therefore, women must be conscious of their medical, social, and sexual behaviour [9]. 

One such social behaviour is smoking, which has been shown to be a cofactor in cervical cancer [16]. Our study showed that only 6.5% of the patients smoked. This low percentage can be because approximately only 11% of the female population in Trinidad smokes [17].

The length of oral contraceptive usage was not specified and stated in participant data collection; therefore, no relationship was found between its use and premalignant cervical lesions. However, most women indicated no use of contraception, increasing their susceptibility to acquiring sexually transmitted infections, such as HIV and HPV. One study (Moreno et al., 2002) found that long-term use of oral contraceptives increased the risk of cervical cancer up to four-fold [18]. In a systematic review performed by Smitha et al. (2020), it was concluded that the longer the use of oral contraceptives, the greater the risk of developing cervical cancer. Additionally, the risk among specific histologic types of cervical carcinoma was evaluated separately, and it was found that the risk was higher for those with adenocarcinoma compared to squamous cell carcinoma [19]. A study performed by Parazzini et al. (1998) suggests that oral contraceptives may have a late-stage (promoter) effect on cervical carcinogenesis and, thus, have public health implications since the incidence of invasive cervical cancers is low at young ages when oral contraceptive use is more common and increases during middle age [20]. Another study by Parazzini et al. (1989) found that the risk of intraepithelial neoplasia was lowered with the use of barrier methods of contraception [21].

Boyles (2007) noted that the use of oral contraceptives for at least five years doubled the incidence of cervical cancer [22]. However, this can be viewed as a double-edged sword because, while high parity in women was observed to be a risk factor, taking oral contraceptives to decrease parity was also a risk factor. In our study, the average number of live births was two; this could not conclusively link increasing parity rates in women to the development of premalignant lesions.

In this study, it was seen that most of our patients had coitarche at 20 years of age. Early onset of coitus can lead to increased exposure to HPV, an increased number of sexual partners in one’s lifetime, and a higher likelihood of pregnancy. A comparison study (Louie KS, 2009) showed that females with coitarche before 16 years of age were approximately two times more likely to get invasive cervical cancer than those who initiated sexual intercourse after age 21 [23]. This link was further strengthened among parous women than those who were nulliparous. In a systematic review, Shepherd et al. (2000) also noted that women who initiated sexual intercourse at or around age 15 were twice as likely to contract HPV than those who commenced after 20 years of age [24]. 

Most women had normal appearances of the cervix, vagina, and vulva, no discharge, and reported normal bleeding, which illustrated the asymptomatic nature in the early stages of cervical cancer. 

The pap smear findings in our study revealed a proportional predominance of LSIL, HSIL, and ASCUS. Immunocompetent women with LSIL should be tested for HPV DNA (Deoxyribonucleic Acid) at 12 months. If HPV DNA is positive, it should be managed appropriately and followed up for spontaneous regression or severe progression into more atypical forms (ASCUS, ASC-H, HSIL, SCC) [25]. If repeated smears at 12 months show abnormal findings, they should be referred for colposcopic biopsy, as they are at a higher risk of progression into more atypical or malignant forms [26]. Patients who had LSIL with previous normal smears are advised for a follow-up smear at 6–12 months and are managed according to the pap smear results in the follow-up test. If the follow-up smear shows ASCUS or a higher grade of atypia, colposcopy is advised; if the result is normal, they are advised to return for annual screening [27].

Cytological findings except HSIL cannot stand alone to confirm cervical cancer. They inform the gynaecologist that there is a presence of abnormal cells, and such abnormalities must undergo biopsies or colposcopies for diagnosis [28].

Of the 33 patients with histopathological testing, 12 had CIN I, which develops slowly; most cases regress independently. Treatment is not required, but rather, the condition will be placed under observation. Ten patients had CIN II, and eight had CIN III, indicating they possess high progression risks; therefore, treatment was necessary. However, studies have shown that 50% of CIN II cases regressed on their own [29]. This information echoes and magnifies the importance of early screening, which can lead to early detection and treatment.

Locally, there are free cervical screening programs in the public health sector; these include cytology, HPV DNA testing, and visual inspection of acetic acid (VIA). Although HPV DNA testing has advantages over cytology and VIA, cytology continues to be mainly used. The target population, 25–49 years, in 2009 and 2010 comprised 11% and 13.2%, respectively, regarding the total number of pap smears performed [4]. It was found that more than 50% of these patients never sought follow-up visits as recommended for a repeat pap smear. This showed that the Regional Health Authority did not achieve adequate coverage for women with abnormal pap smears in 2009 and 2010 [4].

In 2013, Trinidad and Tobago’s Ministry of Health expanded the National Immunisation Programme by introducing human papillomavirus vaccination (HPV). The communication strategy plan included conducting first-sensitisation sessions with media personnel, HPV vaccination promotional posters and brochures, and numerous training and sensitisation sessions for healthcare workers, primary and secondary school staff, parents, and religious groups.

The HPV vaccination of pre-adolescent girls is delivered as a school-based program. Currently, HPV vaccinations are performed free of charge by appointment at all health centres on days scheduled for immunisation, and there are also vaccination and pap smear drives [30].

Sealy et al. (2020) explored barriers and facilitators that affected the acceptance of the HPV vaccine by mothers of adolescents in Trinidad and Tobago. Three major themes emerged: (a) cervical cancer and vaccine knowledge, (b) barriers to uptake, and (c) rephrasing the vaccine strategy. Data indicated that overall strategies to educate the population about the vaccine had yet to occur. Barriers to vaccine uptake were related to a lack of information on the efficacy and safety of HPV vaccines. Parents were unaware that HPV caused cervical cancer. Hence, the study proposed that physicians and other health professionals be used to deliver targeted messages to parents and adolescents to improve vaccine uptake [31].

Compared to other Caribbean countries, Trinidad and Tobago were noted for having more human and infrastructural resources with a population-based cancer registry. However, it lacks vital components, such as a program evaluation, a cervical cancer policy, a program manager, and a budget for the program. There is also no information system for follow-up and no civil society organisation to cooperate in cervical cancer communication, social mobilisation, and advocacy [32].

Human papillomavirus vaccination may reduce the cervical cancer burden through primary prevention of HPV infection. Two brands of HPV vaccines are available: a bivalent (Ceravix) and a quadrivalent vaccine that is also effective against non-oncogenic types, HPV 16 and 11, which cause the most genital warts [33]. Human papillomavirus infections are usually acquired soon after sexual debut; therefore, the vaccine is most effective if administered before the onset of sexual activity. The HPV vaccine is routinely recommended for 11–12-year-old girls and 13–26-year-old females [33].

The relevance and incidence of cancers in Trinidad and Tobago are principally related to reproductive organs in women, namely, breast, cervical, and uterine cancers, and prostate, lung, and colorectal cancers among men. Average incidence rates were highest in areas covered by the Tobago Regional Health Authority (TRHA) (188 per 100,000), while average mortality rates were highest in areas covered by the North West Regional Health Authority (108 per 100,000). Nationals of African ancestry exhibited the highest rates of cancer incidence (243 per 100,000) and mortality (156 per 100,000) compared to their counterparts who were of East Indian descent (incidence, 125 per 100,000; mortality, 66 per 100,000) or mixed ancestry (incidence, 119 per 100,000; mortality, 66 per 100,000) [34,35,36,37].

## 5. Limitations

The major limitation encountered was the lack of availability of data in The National Cancer Registry for the years after 2011. Additionally, there is no formal screening registry in Trinidad and Tobago, meaning data had to be collected from the Cytology Lab’s logbook. Only 33 traceable histopathological reports were found in the RHA records. 

If more than one regional health authority had been used, the sample population would have had more diversity, and the value of data trends would have been strengthened. 

Our sample population included patients from health centres under NCRHA, and cytology request forms varied among the different health centres. Forms that excluded important information, such as the number of sexual partners and the first age of coitus, had to be omitted, hence limiting our sample size.

## 6. Conclusions

The significant risk factors observed for cytology abnormalities and premalignant lesions were early age of coitus, a high number of sexual partners, and no use of contraception, and patients mostly presented as asymptomatic despite obtaining abnormal pap smear results. This reiterates the importance of pap smear tests as a screening tool for cervical cancer in all eligible females.

## Figures and Tables

**Table 1 vaccines-11-00697-t001:** Frequency table showing a summary of the socio-demographic characteristics of the sample population.

Demographic Characteristics		Frequency	Percentage (%)
Age Group (years)	18–25	30	14.0
	26–33	60	27.9
	34–41	66	30.7
	42–49	38	17.7
	50–57	14	6.5
	58–65	5	2.3
	Over 65	2	0.9
Total		215	100.0
Ethnicity	African	136	63.3
	Indian	33	15.3
	Mixed	46	21.4
Total		215	100.0
Marital Status	Single	92	42.8
	Married	50	23.3
	Separated	5	2.3
	Divorced	15	7.0
	Widowed	6	2.8
	Common Law	44	20.5
	Unspecified	3	1.4
Total		215	100.0
Location	Central	31	14.4
	East	98	45.6
	North	77	35.8
	South	3	1.4
	West	6	2.8
Total		215	100.0

**Table 2 vaccines-11-00697-t002:** Measurement of central tendencies, standard deviation, and minimum and maximum for numeric risk factors.

Measure of Central Tendencies:	Age	First Age of Coitus	Number of Sexual Partners	Number of Pregnancies	Number of Live Births
Mean	36.7	18.1	3.7	2.9	2.0
Median	36.0	18.0	3.0	3.0	2.0
Mode	31.0	18.0	1.0	0.0	0.0
Std. Deviation	10.1	3.1	3.4	2.4	1.7
Minimum	18.0	10.0	1.0	0.0	0.0
Maximum	81.0	30.0	23.0	13.0	8.0

**Table 3 vaccines-11-00697-t003:** Showing the sample population who had a family history of cancer.

Family History of Cancer	Frequency	Percentage (%)
Has a family history of cancer	82	38.5
Does not have a family history of cancer	133	61.5
Total	215	100.0

**Table 4 vaccines-11-00697-t004:** Use of contraception in the sample population.

Use of Contraception:	Frequency	Percentage (%)
Uses contraception	82	38.1%
Does not use contraception	133	61.9%
Total	215	100.0

**Table 5 vaccines-11-00697-t005:** Frequency table showing a summary of clinical findings of the sample population.

Clinical Characteristics	Results	Frequency	Percentage (%)
Bleeding	Absent	177	82.3
	Present	38	17.7
Total		215	100.0
Appearance of Vagina	Normal	214	99.5
	Abnormal	1	0.5
Total		215	100.0
Appearance of Vulva	Normal	213	99.1
	Abnormal	2	0.9
Total		215	100.0
Appearance of Cervix	Normal	170	79.1
	Contact Bleed	36	16.7
	Erosion	3	1.4
	Chronic Cervicitis	3	1.4
	Polyp	3	1.4
Total		215	100.0
Presence of Discharge	None	147	68.4
	Purulent	1	0.5
	Leukorrhea	25	11.6
	Frank Blood	5	2.3
	Serosanguineous	2	0.9
	Other	35	16.3
Total		215	100.0

**Table 6 vaccines-11-00697-t006:** Frequency of cytology findings from pap smears in the sample population. All samples are valid.

Results	Frequency	Percentage (%)
ASCUS	59	27.4
HPV	11	5.1
Atypical Glandular Cell	7	3.3
LSIL	70	32.6
ASC-H	5	2.3
HSIL	62	28.8
Invasive Squamous Cell Carcinoma	1	0.5
Adenocarcinoma	0	0
Total	215	100

**Table 7 vaccines-11-00697-t007:** Frequency showing the histopathological findings from patients’ cervical biopsies.

Grade of Cervical Intraepithelial Lesion	Frequency	Percentage
CIN I	12	36.4
CIN II	11	33.3
CIN III	9	27.3
Invasive Squamous Cell Carcinoma reported	1	3.0
Total	33	100%

## Data Availability

The datasets generated and/or analysed during the current study are not publicly available to preserve patients’ confidentialities but are available from the corresponding author on reasonable request.

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
