# Peer review of "The Correlation of Papanicolaou Smears and Clinical Features to Identify the Common Risk Factors for Cervical Cancer: A Retrospective and Descriptive Study from a Tertiary Care Hospital in Trinidad"

_vaccines, 2023, doi:10.3390/vaccines11030697_

Round 1

Reviewer 1 Report

Authors are trying to describe the Correlation of Papanicolaou Smears and Clinical Features to Identify the Common Risk Factors for Cervical cancer: A Retrospective and Descriptive Study from a Tertiary Care Hospital in Trinidad.

However, no logic has been established to satisfy the title. Also, the data are scattered. The logic should be structured with more focus and data. Schematic diagrams should also be utilized for the benefit of the reader.

Author Response

Authors reply: The entire manuscript has been revised and formatted with extensive English language editing. The data has been re-structured and streamlined for better reading and understanding the present study.

Reviewer 2 Report

Keeping in mind the importance to early detect precancerous lesions of cervix, authors of this study  assessed the risk factors for cervical cancer in women from Trinidad and Tobago, correlating the Papanicolaou  smears with clinical feauteres.

Overhall, the manuscript is well done; however, I have some concerns about it.

In Abstract section, authors are are a bit ripetitive; see line 14-15 and 21-22.

Background and Methods are quite clear.

I have also a comment on the Results section: in pages 3 and 5 the authors say: “ as seen in table… (end of manuscript) line 100 and 128 respectively. Why? I would suggest them to remove the frase into the brackets. Tables  could be simplified and their legends have to be improved.

Discussion section is exhaustive.

It would be useful to have the manuscript checked by an English language expert and then, it could be suitable for the publication.

Author Response

Authors reply: Thank you very much for appreciating our hardwork. The entire manuscript has been revised and formatted with extensive English language editing. The data has been re-structured and streamlined for better reading and understanding the present study. The mentioned statements in the abstract and results section has been corrected as per reviewers comments.

Reviewer 3 Report

The authors reported that women in Trinidad who had abnormal pap smear results, majority of them were younger than 42 years old. Geographically, most of them were hailed from Eastern and Northern Trinidad. Major risk factors that appeared to be associated with the abnormal pep smear results were early age of coitus, multiple sexual partners, and no contraceptive use.

Study participants were questioned or observed in a medical setting which is the strength of the study. The study results can be used to initiate an implementation to improve cervical cancer screening services cost-effectiveness. In addition, the study results can be used to generate hypothesis for evidence-based research.

Minor comment:      

-Table 5 Clinical characteristic - using the term abnormal bleeding is a little bit confusing because abnormal bleeding cannot be normal. The authors may consider removing the word abnormal. 

- Be consistent with the decimal places of the numbers reported and format and size of fonts.

Author Response

Authors reply: Thank you for appreciating our hard work.The entire manuscript has been revised and formatted with extensive English language editing. The data has been re-structured and streamlined for better reading and understanding the present study. The mentioned statements in the tables section has been corrected as per reviewers comments. The decimals are corrected and numbers are reported to maintain a uniform format and font size.

Round 2

Reviewer 1 Report

Although it appears to have been completely revised, in reality it has not been revised very much.

No logic has been established to satisfy the title. Also, the data are scattered. The point I made last time that the logic should be structured with a narrower focus and data is also not addressed. Also, there are no additional schematic figures, even though I pointed out that schematic figures should also be utilized for the benefit of the reader.

The authors' response is completely out of touch.

Author Response

The comments addressed by the reviewer is subjected to change the entire presentation of our manuscript thereby that affects the authenticity  of our research. The logic, schematics and data cannot be changed as per the reviewer at this point as this particular research was carried out following proper ethical mode. After all the hard work if the reviewer suggests such major and drastic changes then the validity of our paper would be lost.